# Stem cell-specific NF-κB is required for stem cell survival and epithelial regeneration upon intestinal damage

Aurélia Joly[1,*], Meghan Ferguson[1,2,*], Minjeong Shin[1,2] and Edan Foley[1,2,‡]

## ABSTRACT

Immune signals coordinate the repair of damaged epithelia by intestinal stem cells. However, it is unclear if immune pathways act autonomously within the stem cell to direct the damage response pathway. We consider this an important question, as stem cell dynamics are essential for formation and maintenance of the entire epithelium. We used *Drosophila* to determine the impact of stem cell-specific loss of NF-κB on tissue regeneration upon chemical injury. We found that loss of NF-κB enhanced cell death, impaired enterocyte renewal and increased mortality. Mechanistically, we showed that inhibition of stem cell apoptosis is essential for NF-κB-dependent maintenance of cell viability and tissue repair. Combined, our data demonstrate that stem cell-intrinsic NF-κB activity is essential for an orderly repair of damaged intestinal epithelia.

KEY WORDS: Stem cells, Intestine, Repair, Immunity, IMD

## INTRODUCTION

Intestinal stem cells (ISCs) generate epithelial cells that extract luminal nutrients, present a barrier to noxious agents, and regulate host-microbe interactions. Immune pathways modify epithelial stress responses, and errant immune signals underpin inflammatory illnesses that elevate the risk of colorectal cancer (Ferguson and Foley, 2021; Ullman and Itzkowitz, 2011). For example, defects in the NOD2 bacterial sensor are the greatest single genetic risk factor for Crohn's disease (Hugot et al., 1996, 2001; Ogura et al., 2001; Jostins et al., 2012). As ISCs are critical for maintenance of the epithelial barrier, we consider it important to resolve the extent to which immune signals impact ISC function.

*Drosophila* is an excellent model of intestinal epithelial cell dynamics (Miguel-Aliaga et al., 2018; Buchon et al., 2013; Ludington and Ja, 2020). Like vertebrates, the fly gut is lined by an epithelial layer that is maintained by multipotent ISCs (Micchelli and Perrimon, 2006; Ohlstein and Spradling, 2006). In flies and vertebrates, ISCs typically divide asymmetrically to generate a niche-resident ISC, and an undifferentiated progenitor that migrates apically (O'Brien et al., 2011; Goulas et al., 2012; Perdigoto et al., 2011; Joly and Rousset, 2020),

where it matures as specialist enterocyte or enteroendocrine cells (Ohlstein and Spradling, 2007). In flies, the enterocyte precursor is classified as an enteroblast, and the stem cell-enteroblast pair constitutes the midgut progenitor compartment. Aside from compositional similarities, ISC fates are governed by related signals in flies and vertebrates. In both cases, EGF, JAK-STAT and BMP pathways control ISC proliferation, while Notch promotes enterocyte development (Perdigoto et al., 2011; Ohlstein and Spradling, 2007; Ngo et al., 2020; Buchon et al., 2009b; Bardin et al., 2010; Sallé et al., 2017; Cordero et al., 2012; Nászai et al., 2021; Jin et al., 2015; Biteau and Jasper, 2011; Jiang et al., 2009; Tian and Jiang, 2014; Guo et al., 2013; Takashima et al., 2011). Given the similarities between fly and vertebrate intestinal epithelial organization, *Drosophila* has considerable potential to uncover foundational relationships between ISC immunity and function.

The *Drosophila* immune deficiency (IMD) pathway, an antibacterial response with overt similarities to the mammalian NOD2 pathway, is a prominent regulator of fly responses to gut microbes (Erkosar et al., 2014; Broderick et al., 2014; Buchon et al., 2009b). Detection of bacterial peptidoglycan engages the Imd protein to activate fly IKK and the NF-κB ortholog Relish (Rel) (Myllymäki et al., 2014). Active Rel modifies expression of antimicrobial peptides, stress response pathway genes, and genes that control metabolic activity (Buchon et al., 2009b). Recent work uncovered roles for the IMD-Rel axis in mature epithelial cells (Shin et al., 2022; Liu et al., 2022). In contrast, we know little about ISC-intrinsic roles for IMD in response to acute stresses despite evidence that IMD components are expressed at relatively high levels in the progenitor compartment (Shin et al., 2022; Hung et al., 2020).

We discovered that dextran sodium sulfate-dependent damage activates Rel in stem cells, promoting ISC survival and generation of transient enteroblasts. Inactivation of Rel in ISCs resulted in death of damaged stem cells, a failure to repair damaged tissue, and substantially impaired survival of challenged flies. Our work uncovers a stem cell-intrinsic requirement for an Rel-dependent damage response in the gut progenitor compartment.

## RESULTS

### NF-κB/Rel regulates ISC dynamics

We showed that genetic inactivation of IMD within the female midgut progenitor compartment impairs homeostatic ISC function (Shin et al., 2022). As IMD acts through the JNK and NF-κB/Rel pathways (Boutros et al., 2002), we asked what effects inhibition of *Rel* alone has on ISC dynamics. To characterize effects of Rel deficiency, we used the temperature controlled *esg^{ts}* driver line to express a validated *Rel* RNAi (Vandehoef et al., 2020) construct in midgut progenitors (*esg^{ts}/Rel^{RNAi}*).

As gut architecture deteriorates with age, we compared intestinal physiology in young and old wild-type and *esg^{ts}/Rel^{RNAi}* flies. We did not observe differences between *esg^{ts}/+* and *esg^{ts}/Rel^{RNAi}* flies at early stages of adulthood. In both cases, the gut contained an orderly

[1]Department of Medical Microbiology and Immunology, Faculty of Medicine and Dentistry, University of Alberta, Edmonton, T6G 2R3, AB, Canada. [2]Department of Cell Biology, Faculty of Medicine and Dentistry, University of Alberta, Edmonton, T6G 2R3, AB, Canada.

*Lead authors made equal contributions to this manuscript. Author order was determined randomly using the Siri function on E.F.'s iPhone.

‡Author for correspondence (efoley@ualberta.ca)

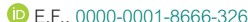 E.F., 0000-0001-8666-3267

Biology Open

epithelium of evenly spaced progenitors among mature enterocyte and enteroendocrine cells (Fig. 1A), as well as equal incidences of ISC proliferation (Fig. 1C). Matching prior studies (Rodriguez-Fernandez et al., 2020), 30-day-old wild-type intestines displayed age-linked dysplasia that included irregular epithelial spacing (Fig. 1B), and elevated rates of ISC proliferation (Fig. 2C). In contrast, progenitor-restricted loss of *Rel* prevented age-dependent decline of epithelial organization (Fig. 1B) and resulted in significantly fewer ISC mitoses (Fig. 1C). Progenitor-specific knockdown of the IKKγ ortholog *Kenny* (*key*) also prevented age-dependent increases in ISC mitosis (Fig. 1D), confirming a role for the IKK/NF-κB axis in age-associated deterioration of ISC function.

As progenitor-specific loss of Rel affected epithelial organization in older flies, we quantified the impact of *Rel* depletion on the number and identity of midgut progenitors in 30-day-old flies. Consistent with links between progenitor cell Rel activity and ISC proliferation, we observed significantly fewer midgut progenitors in $esg^{ts}/Rel^{RNAi}$ flies relative to $esg^{ts}/+$ counterparts (Fig. 1E). To test if Rel influences progenitor cell identity, we quantified the ISC to enteroblast ratio in midguts of $esg^{ts}/Rel^{RNAi}$ flies and $esg^{ts}/+$ controls. To do so, we employed a line that allows progenitor-specific depletion of *Rel* while marking enteroblasts with CFP and GFP reporters, and ISCs with CFP alone (Martin et al., 2018) ($esgGAL4$ $UAS$-$CFP$, $Su(H){:}GFP/UAS$-$Rel^{RNAi}$; $GAL80^{ts}$, Fig. 1F). In wild-type intestines ($esgGAL4$ $UAS$-$CFP$, $Su(H){:}GFP/+$; $GAL80^{ts}$), roughly 50% of all progenitors expressed enteroblast markers, suggesting equal numbers of ISC and enteroblasts (Fig. 2G). In contrast, only 40% of *Rel*-deficient progenitors expressed enteroblast markers (Fig. 2F,G), indicating that loss of Rel

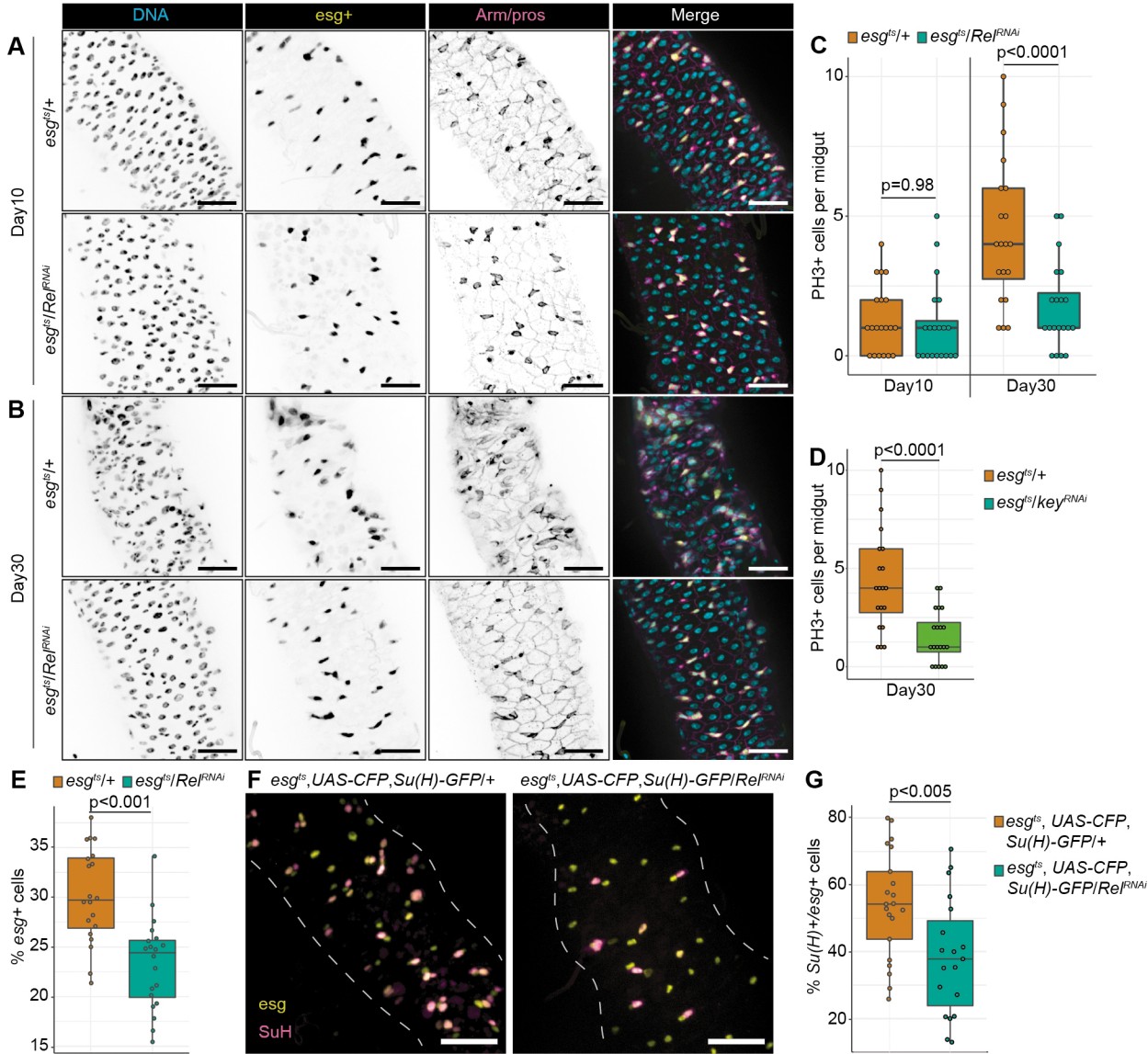

**Fig. 1. NF-κB regulates intestinal stem cell proliferation.** (A) Posterior midguts of 10-day-old $esg^{ts}/Rel^{RNAi}$ and $esg^{ts}/+$ flies, with DNA in cyan, progenitors in yellow, enteroendocrine cells in magenta, and cell borders labelled by Armadillo in magenta. (B) Posterior midgut after 30 days of *Rel* depletion. (C) PH3+ mitotic cells in $esg^{ts}/Rel^{RNAi}$ and $esg^{ts}/+$ midguts. Significance calculated using ANOVA followed by pairwise Tukey tests. (D) PH3+ cells after progenitor-specific knockdown of IKKγ homolog *kenny* (*key*). (E) Percentage of intestinal epithelial cells that are GFP+ in 30-day old $esg^{ts}/Rel^{RNAi}$ and $esg^{ts}/+$ midguts. (F) $esg^{ts}$, $UAS$-$CFP$, $Su(H)$-$GFP$ flies after 30 days of progenitor-specific *Rel* depletion with esg in yellow and Su(H)+ enterocyte precursors in magenta. (G) Proportion of Su(H)+ enterocyte precursors within the indicated progenitor pools. For D, E and G significance found using Student's *t*-test. Scale bars: 25 µm.

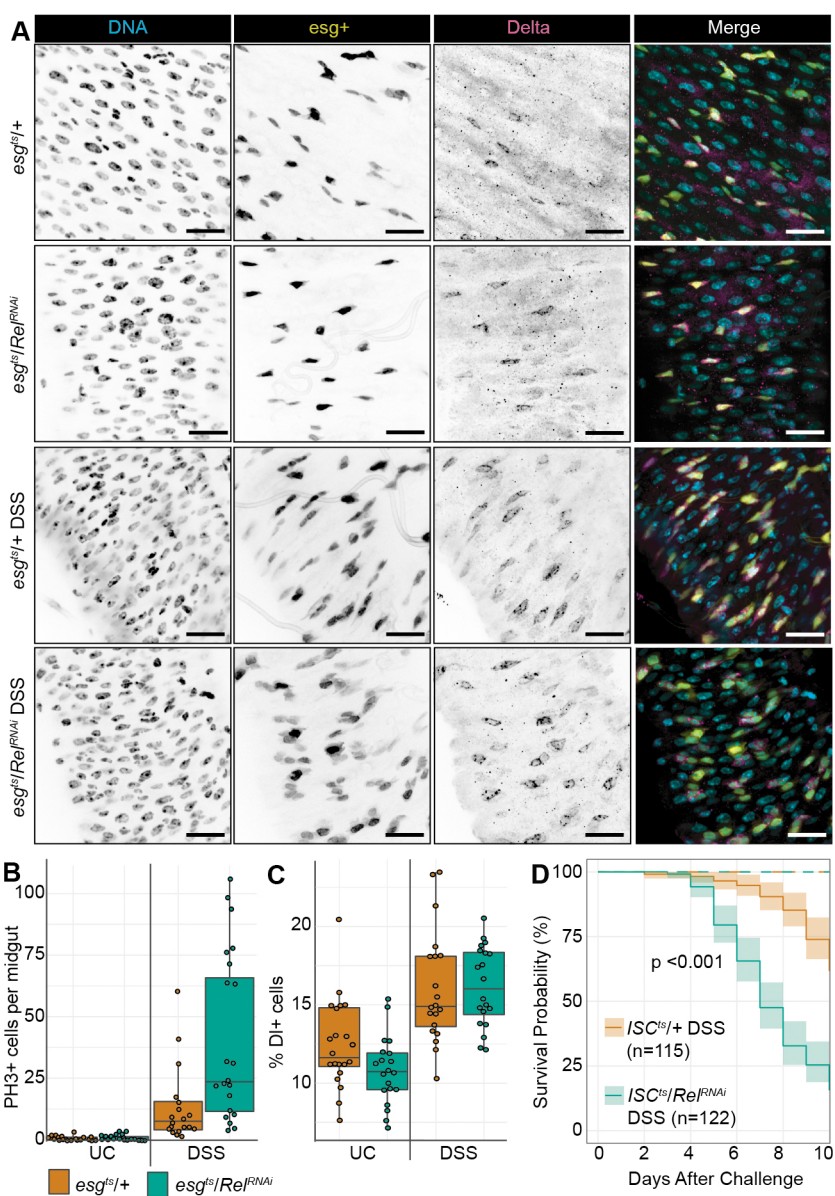

**Fig. 2. Rel is not essential for DSS-dependent ISC proliferation but impacts survival rates after DSS treatment.** (A) 10-day-old unchallenged (UC) $esg^{ts}/+$ flies fed PBS/5% sucrose solution for 48 h; and DSS-treated $esg^{ts}/+$ or $esg^{ts}/Rel^{RNAi}$ flies fed 3% DSS in 5% sucrose for 48 h. Scale bars: 25 µm. DNA (cyan), esg+ progenitors (yellow), and Dl+ ISCs (magenta) labelled. (B) PH3+ cells of 48 h UC and DSS treated $esg^{ts}/+$ midguts. (C) Percentage Dl+ cells in UC or DSS-treated $esg^{ts}/+$ or $esg^{ts}/rel^{RNAi}$ flies. (D) Survival of $ISC^{ts}/+$ and $ISC^{ts}/Rel^{RNAi}$ flies treated with DSS (solid lines), or with a sucrose control (dashed lines). Significance calculated with a log rank test.

impairs enteroblast generation. Combined, our data implicate intestinal Rel activity in progenitor cell proliferation.

## NF-κB promotes epithelial regeneration

As progenitors renew damaged epithelia, we expanded our work to determine if progenitor cell Rel activity affects epithelial regeneration after enteric stress. Specifically, we characterized progenitor dynamics in $esg^{ts}/+$ and $esg^{ts}/Rel^{RNAi}$ flies that we fed dextran sodium sulfate (DSS), a damaging polysaccharide that promotes a rapid regenerative response (Amcheslavsky et al., 2009; Ren et al., 2010). As expected, DSS disrupted the organization of intestinal epithelial cells in $esg^{ts}/+$ and $esg^{ts}/Rel^{RNAi}$ midguts (Fig. 2A). Although highly variable from animal to animal, loss of Rel did not impede ISC proliferation (Fig. 2B), or the generation of midgut Delta+ ISCs (Fig. 2C), indicating an intact proliferative response in Rel-deficient progenitors.

Since the *esg* driver allows for the expression of transgenes in both ISCs and enteroblasts, we wanted to focus on roles of Rel specifically in ISCs. To do this we employed the $ISC^{ts}$ fly line ($w$ ; $esg$-$GAL4,UAS$-$2xEYFP;Su(H)GBE$-$GAL80,tub$-$GAL80^{ts}$) to

knock down *Rel* specifically in ISCs, then monitored fly survival in response to DSS. We found that depletion of Rel from ISCs significantly impaired the ability of flies to survive DSS treatment. By day ten, more than 85% of $ISC^{ts}/Rel^{RNAi}$ flies died, whereas only 40% of $ISC^{ts}/+$ flies had died (Fig. 2D), suggesting that $ISC^{ts}$-mediated depletion of Rel impacts the ability of adult flies to survive challenges with DSS. As a caveat, we note that we cannot formally exclude the possibility that the $ISC^{ts}/Rel^{RNAi}$ fly line also impacts *Rel* expression in early enteroblasts.

To determine if Rel affects epithelial regeneration after a DSS challenge, we used the $esg^{ts}$ Flip-Out ($esg^{ts}$ F/O) lineage tracing system to track epithelial renewal in damaged intestines. The $esg^{ts}$ F/O system marks *esg*-positive progenitors and their offspring with GFP in the adult midgut (Jiang et al., 2009). As anticipated, unchallenged flies generated a limited number of GFP+ wild-type ($esg^{ts}$ F/O/+) or Rel-deficient ($esg^{ts}$ F/O/ $Rel^{RNAi}$) clones (Fig. 3A). In contrast, whereas wild-type progenitors generated the expected large GFP-marked clones during a regenerative response (Fig. 3A,B), *Rel*-deficient progenitors produced very few clones that were markedly reduced in size (Fig. 3A,B). Thus, progenitor-specific *Rel*

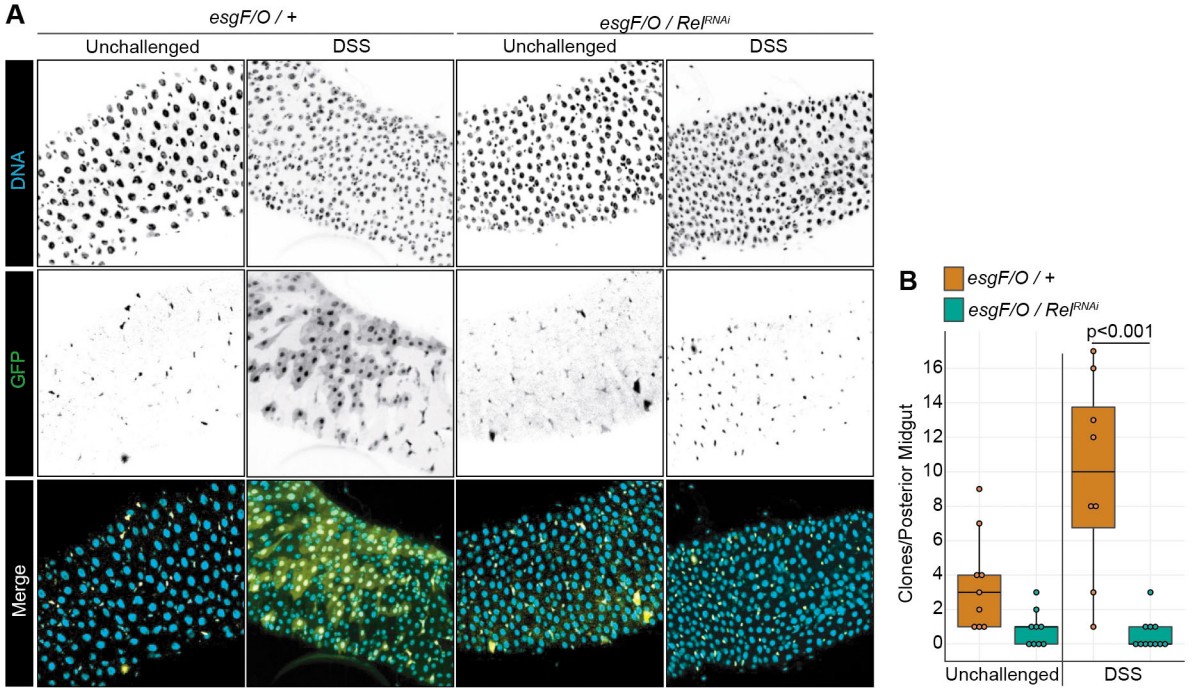

**Fig. 3. Rel is essential for epithelial regeneration.** (A) DNA (cyan) and GFP-marked clones (yellow) generated by unchallenged or DSS-treated wild-type (*esg FO/+*) or Rel-deficient (*esg FO/Rel^RNAi*) progenitors. We induced flip-out clones by incubating flies at 29°C, followed by 48 h treatment with DSS (passed into fresh vials after 24 h), with an additional 5 days of recovery at 25°C to allow daughter cell regeneration. (B) Quantification of clones of three or more cells in regions R4 or R5 of the posterior midguts of flies with the indicated genotypes and treatments. Significance calculated using ANOVA followed by pairwise Tukey tests.

activity appears dispensable for ISC proliferation, but essential for epithelial renewal after DSS-dependent enteric stress.

## NF-κB supports ISC survival during stress

To understand the involvement of ISC-specific Rel function in epithelial renewal, we performed single-cell gene expression analysis on midguts dissected from unchallenged, or DSS-treated, *ISC^ts/+* and *ISC^ts/Rel^RNAi* flies. For each group, we generated profiles of roughly 2500 cells. For example, control *ISC^ts/+* flies yielded 2589 cells in seven transcriptional clusters (Fig. 4A) that expressed markers (Hung et al., 2020) of progenitors; pre-enteroendocrine cells; enteroendocrine cells; anterior, middle, and posterior enterocytes; and copper cells (Fig. 4B). We resolved two clusters each from anterior, middle, and posterior enterocytes (Fig. 4C-E), and seven enteroendocrine clusters (Fig. 4F), including one that expressed the *esg* progenitor marker, indicating that it corresponded to the committed enteroendocrine precursor (Fig. 4G). The remaining six clusters had non-overlapping hormone profiles, suggesting that each cluster corresponds to an enteroendocrine subtype (Fig. 4H).

In agreement with Fig. 1, cellular identities within unchallenged *ISC^ts/+* and *ISC^ts/Rel^RNAi* midguts were broadly similar (Fig. 5A-C). Inactivation of Rel had anticipated effects on progenitor-specific immune signals (Fig. 5D), confirming the efficacy of the *Rel^RNAi* line. However, the consequences of Rel inactivation matched earlier demonstrations that Rel has far-reaching effects on midgut transcription (Broderick et al., 2014). Inactivation of Rel attenuated expression of genes required for ISC growth, and diminished expression of genes associated with metabolism and detoxification in enterocytes (Fig. 5D). Loss of Rel had profound effects on expression of genes linked with enteroblast development [e.g. *E(spl)mgamma-HLH*, *E(spl)m3-HLH*, *Egfr*, *Dl*, Fig. 5E],

overlapping with the loss of enteroblasts in Rel-deficient progenitors (Fig. 1).

To map impacts of Rel and DSS on the intestinal epithelium, we performed an integrated analysis of gene expression profiles in unchallenged or DSS-treated *ISC^ts/+* flies (Fig. 6A). We found that DSS-treated guts were dominated by a cell type that is rare in unchallenged intestines and is predominately marked by expression of genes associated with metabolism (Fig. 6A-D). As these cells expressed low levels of enterocyte markers (Fig. 6B), and an earlier study showed accumulation of immature enterocytes in DSS-treated flies (Amcheslavsky et al., 2009), we provisionally annotated these cells as pre-enterocytes, although we caution that additional experiments are required to determine their developmental trajectory. DSS engaged growth and homeostatic pathways in progenitors (Fig. 6C), including JAK-STAT and EGF repair pathways, as well as Notch and BMP cell specification responses, pointing to directed epithelial regeneration (Fig. 6D,E). Alongside growth and repair pathways, we discovered that DSS also initiated immune response within progenitors (Fig. 6D), including induction of Rel response genes such as *pirk*, *PGRP-SD*, and *PGRP-LB* (Fig. 6E).

As Rel is essential for regeneration after a DSS challenge (Fig. 3), we then compared expression profiles in DSS-treated *ISC^ts/Rel^RNAi* and *ISC^ts/+* midguts to determine the effects of Rel-deficiency on epithelial regeneration. In agreement with a requirement for Rel in epithelial renewal (Fig. 3), we observed significantly fewer pre-enterocytes in DSS-challenged *ISC^ts/Rel^RNAi* midguts relative to DSS-treated *ISC^ts/+* controls (Fig. 6F), and we found that blocking Rel attenuated progenitor-specific expression of multiple IMD/Rel targets, including *Rel* [e.g. *pirk*, *PGRP-LB*, *PGRP-SC2*, *Rel* (Fig. 6G-I)], further confirming the specificity of the *Rel^RNAi* line. Notably, *Rel* inhibition also inhibited expression of genes linked to gut homeostasis, and cell survival (Fig. 6G,I). For example,

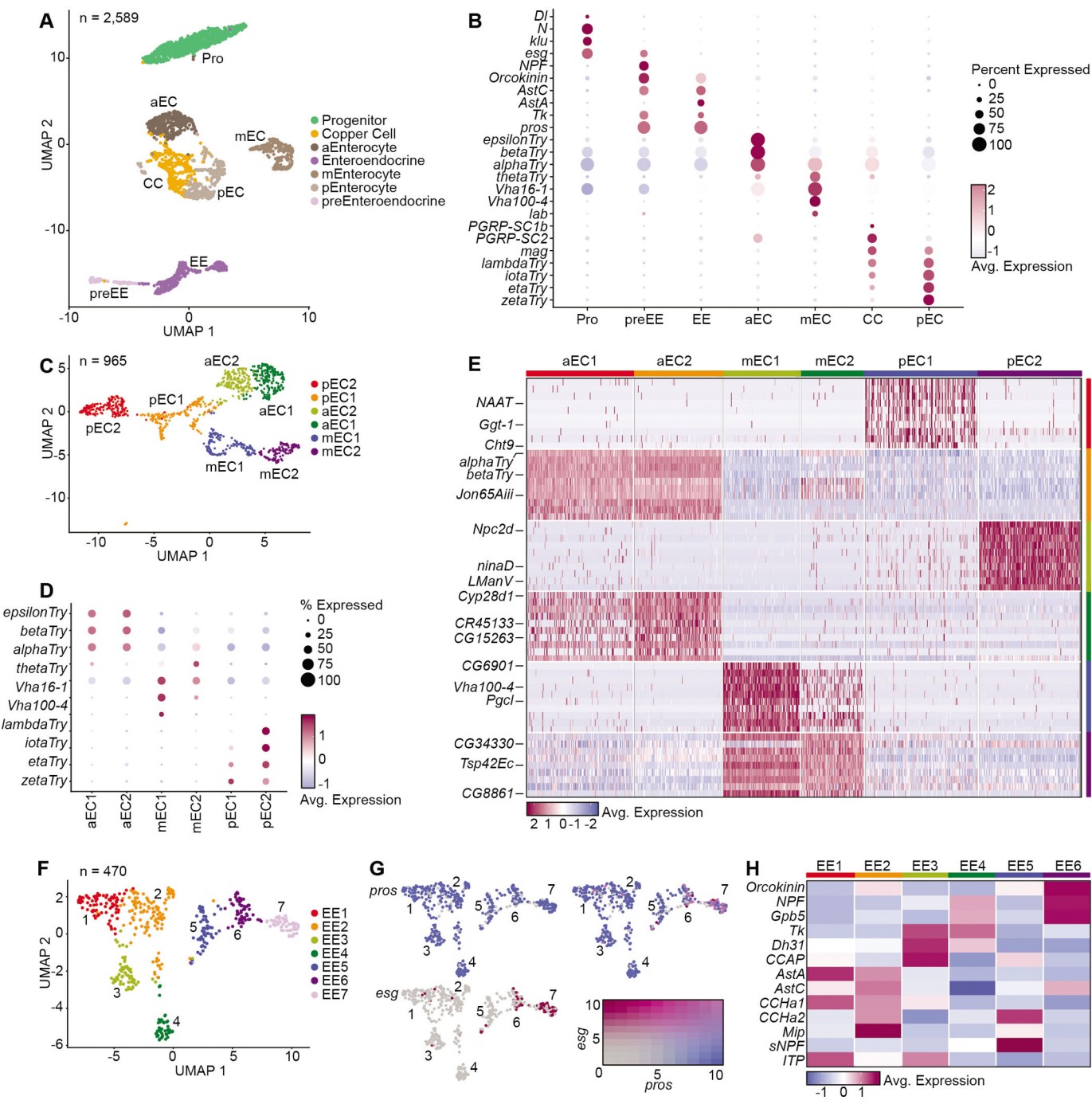

**Fig. 4. Midgut single cell atlas of *ISC^ts/+* intestines.** (A) Cell types in *ISC^ts/+* intestines. (B) Marker expression in the indicated cells. (C) Six enterocyte subtypes. (D) Dotplot of anterior (aEC), middle (mEC), and posterior (pEC) enterocyte marker expression in each cluster. (E) Marker expression heatmap for each cluster. (F,G) Prospero-positive enteroendocrine subtypes (F), with a feature plot (G) showing expression of the progenitor marker *esg* in cluster seven. (H) Heatmap of peptide hormone expression in each enteroendocrine subtype. Pro, progenitor; preEE, pre-enteroendocrine; ee, enteroendocrine; aEC, mEC, and pEC, anterior, middle and posterior enterocytes, respectively; CC, copper cells.

blocking *Rel* attenuated expression of the *p53* and *Apc* tumor suppressor genes, and elevated expression of the pro-apoptotic factors *Dronc* and *Drice* (Fig. 6H).

## Rel promotes stem cell survival during enteric stress

Our data suggest failed epithelial regeneration and altered expression of cell survival genes in DSS-treated guts with Rel-deficient ISCs. To test if *Rel* is necessary for stem cell survival during exposure to DSS, we quantified apoptotic cells in challenged

and unchallenged *ISC^ts/+* and *ISC^ts/Rel^RNAi* flies (Fig. 7A). Quantification of TUNEL+ ISCs suggested a statistically insignificant elevation of ISC apoptosis in *ISC^ts/Rel^RNAi* midguts relative to *ISC^ts/+* controls (Fig. 7B). By contrast, we observed roughly twice as many apoptotic ISCs in DSS-treated *ISC^ts/Rel^RNAi* midguts compared to their DSS-treated *ISC^ts/+* counterparts (Fig. 7A,B), suggesting a role for Rel in prevention of ISC death.

To determine if failed epithelial repair occurs in *ISC^ts/Rel^RNAi* midguts due to excess ISC death, we tracked midgut regeneration in

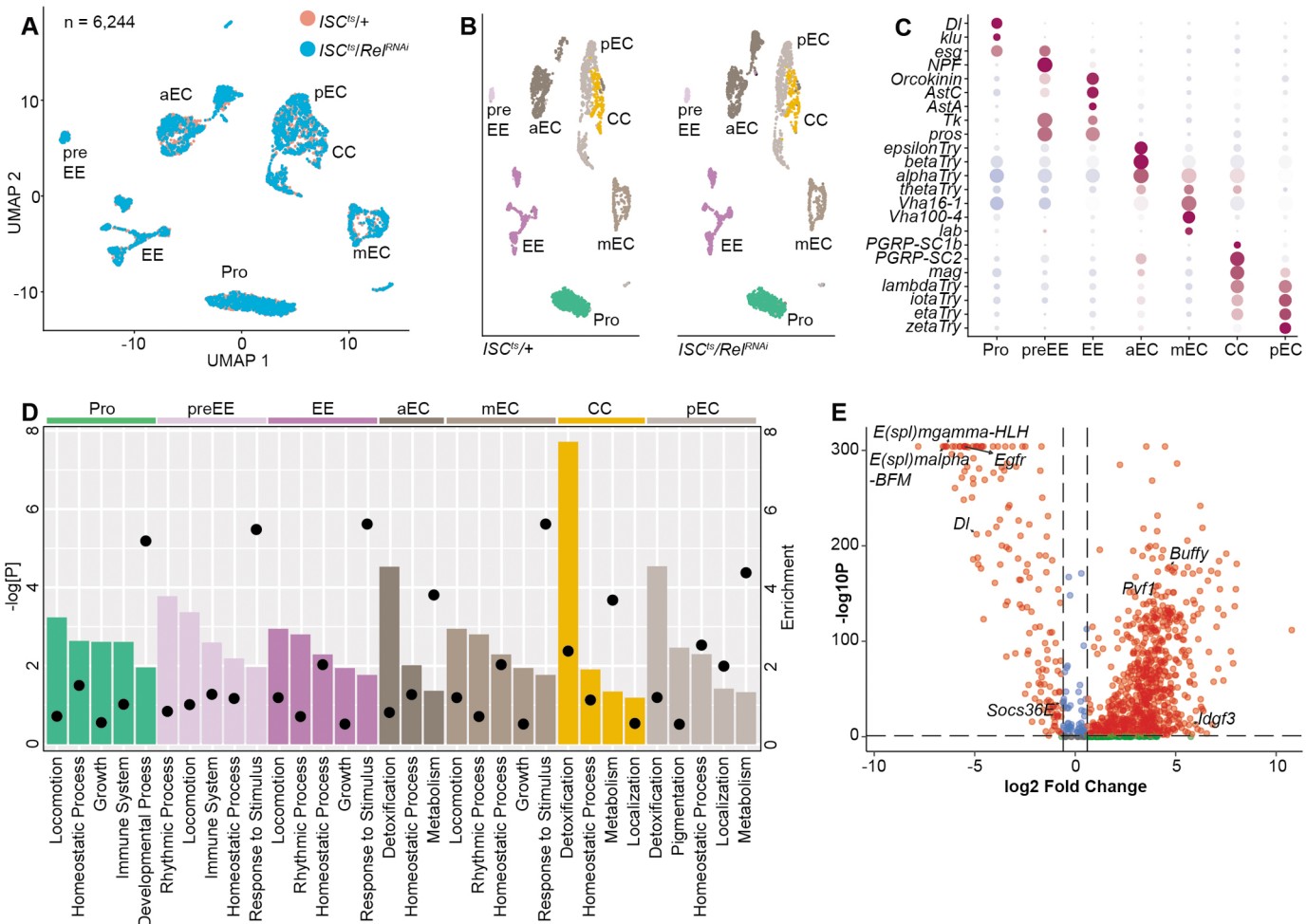

**Fig. 5. Rel impacts epithelial transcription.** (A,B) Integrated (A), and individual (B) UMAP projections of cell types in $ISC^{ts}/+$ and $ISC^{ts}/Rel^{RNAi}$ midguts. (C) Marker expression in each cell type. (D) The top five downregulated GO terms in $ISC^{ts}/Rel^{RNAi}$ midguts relative to $ISC^{ts}/+$ controls. Enrichment is indicated by bar length and negative log $P$-values with circles. (E) Volcano plot showing progenitor-specific gene expression in $ISC^{ts}/Rel^{RNAi}$ midguts relative to $ISC^{ts}/+$ controls. Pro, progenitor; preEE, pre-enteroendocrine; ee, enteroendocrine; aEC, mEC, and pEC, anterior, middle and posterior enterocytes, respectively; CC, copper cells.

control flies ($esg^{ts}$ $F/O/+$); flies that lack $Rel$ within the progenitor compartment ($esg^{ts}$ $F/O/Rel^{RNAi}$); flies that express the anti-apoptotic p35 protein in progenitors ($esg^{ts}$ $F/O/p35$), and flies that express p35 in Rel-deficient progenitors $esg^{ts}$ $F/O/Rel^{RNAi}$, $p35$). These fly lines express the respective transgenes in all GFP-marked progenitors and their mature progeny. Matching our earlier work, we detected large GFP-marked clones specifically in DSS-treated wild-type flies that were absent from DSS-treated $Rel$-deficient flies (Fig. 7C,D). As expected, progenitor-specific expression of p35 increased the number of clones produced by wild-type cells in the absence or presence of a DSS challenge (Fig. 7C,D). Notably, expression of p35 was sufficient to restore epithelial regeneration to DSS-exposed Rel-deficient clones (Fig. 7C,D), indicating that inhibition of apoptosis within Rel-deficient cells restores regenerative capacity to the midgut.

## DISCUSSION

Flies are an ideal animal to scrutinize links between stem cell-specific NF-κB activity and barrier organization (Khan et al., 2023; Tafesh-Edwards and Eleftherianos, 2023; Buchon et al., 2014; Capo et al., 2019). Transcriptomic and physiological studies point to elaborate connections between intestinal IMD pathway activity and cellular

functions (Broderick et al., 2014; Liu et al., 2022; Shin et al., 2022; Zhai et al., 2018; Broderick, 2016; Lesperance and Broderick, 2020; Barretto et al., 2020; Guo et al., 2014; Yamashita et al., 2021; Kosakamoto et al., 2020). Our work aligns with prior studies that IMD regulates age-dependent decay of epithelial organization (Guo et al., 2014). However, our study also characterized effects of Rel on epithelial regeneration after challenges with DSS. We found that DSS induced progenitor cell-specific expression of Rel signature genes, indicating induction Rel activity during enteric stress, and we discovered that ISC-specific loss of $Rel$ led to stress-dependent progenitor death that impaired damage-responsive generation of pre-enterocytes. Our observations implicate IMD in the survival of DSS-exposed stem cells and raises questions that we feel merit consideration in future studies.

How do progenitors detect IMD ligands during periods of stress? Typically, Rel responds to bacterial DAP-type peptidoglycan via transmembrane and intracellular PGRPs (Kaneko et al., 2006, 2004; Leulier et al., 2003; Takehana et al., 2004; Gottar et al., 2002; Choe et al., 2002; Rämet et al., 2002; Choe et al., 2005), although commensal-derived acetate also activates Rel in midgut enteroendocrine cells (Jugder et al., 2021). A dense, chitinous peritrophic matrix combined with a contiguous layer of mature

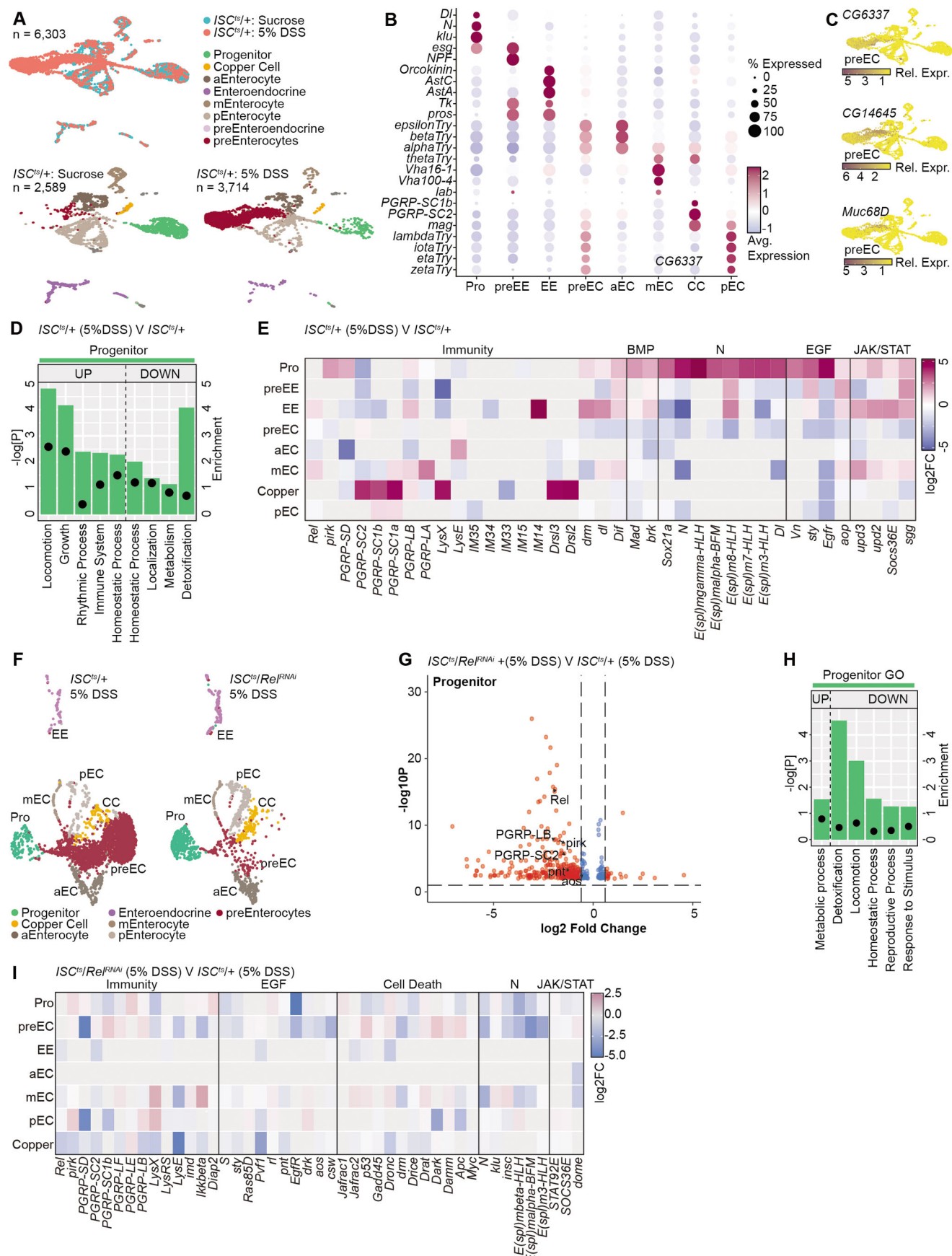

**Fig. 6.** See next page for legend.

**Fig. 6. Rel modifies apoptosis regulators in DSS-treated guts.** (A) UMAP projections of sucrose or DSS-treated midguts. (B) Marker expression in DSS-challenged flies. (C) Top five up-, or downregulated GO terms in DSS-exposed *ISC^ts*/+ progenitors relative to unchallenged controls. Enrichment indicated by bar length and the negative log *P*-values with closed circles. (D) Representative gene expression in DSS-treated *ISC^ts*/+ midguts relative to untreated *ISC^ts*/+ controls. (E) UMAP projections of DSS-treated *ISC^ts*/+ and *ISC^ts*/*Rel^RNAi* midguts. (F) Progenitor-specific expression in DSS-treated *ISC^ts*/*Rel^RNAi* midguts relative to DSS-treated *ISC^ts*/+ controls. (G) The top deregulated GO terms in DSS-exposed *ISC^ts*/*Rel^RNAi* progenitor cells relative to DSS-treated *ISC^ts*/+ controls. Enrichment indicated by bar length and negative log *P*-values with closed circles. (H) Gene expression heatmap in DSS-treated *ISC^ts*/*Rel^RNAi* midguts relative to DSS-treated *ISC^ts*/+ controls.

epithelial cells prevents excess environmental interactions with basolateral ISCs (Galenza et al., 2023). As a result, it is unclear how luminal DSS prompts Rel activation in such a shielded cell. In a simple model, DSS may breach the barrier to the extent that gut-resident bacterial patterns access immune receptors on progenitors. If paracellular leakage of bacterial patterns initiates immune responses in progenitors, it would be of interest to test how defined bacterial communities, including absence of gut bacteria, modify stem cell survival and barrier regeneration after a DSS challenge. With a simple, manipulable microbiome, and straightforward infection protocols, flies are an excellent model to ask how hosts, commensals, and pathogens interact to shape stress-dependent epithelial regeneration (Buchon et al., 2013, 2009a; Lesperance and Broderick, 2020; Barron et al., 2024).

How does Rel modify stem cell survival? As Rel has broad effects on gene expression within the midgut, including genes linked to stress responses and development (Broderick et al., 2014), it is possible that Rel directly modifies expression of genes involved in cell survival. However, we cannot exclude the possibility that Rel controls survival via intermediate pathways. Moving forward, it will be of interest to test if interactions between Rel and EGF, Notch or JAK-STAT are important for DSS-dependent control of stem cell viability. Beyond, EGF, Notch, and JAK-STAT, we feel it will be pertinent to test the tumor suppressor BMP pathway as a potential mediator of Rel-dependent stem cell survival. BMP regulates epithelial patterning (Tian and Jiang, 2014; Boutros et al., 2002; Tracy Cai et al., 2019; Tian et al., 2017; Guo et al., 2013; Driver and Ohlstein, 2014; Zhou et al., 2015; Christensen et al., 2024); DSS activates the BMP pathway in progenitors (Fig. 3D); and *Vibrio cholerae* engages a Rel-BMP axis to blocks ISC proliferation (Xu and Foley, 2024), suggesting possible interactions between Rel and BMP in the regulation of cell viability after exposure to DSS.

Finally, we note that our work focused on the adult female midgut, a common practice within the *Drosophila* community. As there is an abundance of evidence that gut structure, function, immunity and proliferation are sexually dimorphic (Hudry et al., 2016; Regan et al., 2016, 2022; Blackie et al., 2024), we believe it would be of interest to test effects of Rel inactivation on the male ISC response to extrinsic challenges.

## MATERIALS AND METHODS
### *Drosophila* husbandry
*Drosophila* crosses were set up and maintained at 18°C on standard corn meal food (Nutri-Fly Bloomington formulation; Genesse Scientific). All experimental flies were virgin females and kept at a 12 h:12 h light dark cycle throughout. Upon eclosion, flies were kept at 18°C then shifted to the appropriate temperature once 25-30 flies per vial was obtained. Fly lines used in this study were: *w ; esg-GAL4,tubGAL80ts,UAS-GFP* (*esg^ts*) (Jiang et al., 2009), *w ; esg-GAL4,UAS-2xEYFP;Su(H)GBE-GAL80,tub-GAL80^ts* (*ISC^ts*)

(Wang et al., 2014), *w ; esg-GAL4, tub-GAL80^ts, UAS-GFP ; UAS-flp, Act>CD2>GAL4* (*esg F/O*) (Jiang et al., 2009), *w ; esg-GAL4,UAS-CFP, Su(H)-GFP;tubGal80^ts (esgts,UAS-CFP,SuH-GFP)* (Martin et al., 2018), *w^1118* (VDRC #60000), *Rel^RNAi* (VDRC #49413), *key^RNAi* (VDRC #7723), w;;*UAS-p35* (Loudhaief et al., 2017). For clonal analyses, we induced flip-out clones by incubating flies at 29°C, followed by a 48 h treatment with DSS, where flies were transferred to DSS-containing vials for 24, then flipped to fresh DSS-containing vials for an additional 24 h. Afterwards, flies were raised for a 5 day recovery period at 25°C to allow daughter cell regeneration.

### Immunofluorescence
Intestines were dissected in PBS, fixed in 8% formaldehyde for 20 min, washed in PBS 0.2% Triton-X (PBST) then blocked in PBST with 3% BSA for 1 h at room temperature. Primary antibodies were incubated in PBST with BSA overnight at 4°C. The following day guts were washed in PBST then secondary antibody incubations were done in conjunction with DNA stain for 1 h at room temperature in PBST with BSA, washed with PBST then again with PBS. Primary antibodies used: anti-prospero [1/100; Developmental Studies Hybridoma Bank (DSHB) MR1A], anti-armadillo (1/100;DSHB N2 7A1), chicken anti-GFP (1/2000; Invitrogen PA1-9533), anti-phospho-histone3 (1/1000; Millipore 06-570), anti-Delta (1/100; DSHB C594.9B). Secondary antibodies used: goat anti-chicken 488 (1/1000; Invitrogen A11039), goat anti-mouse 568 (1/1000; Invitrogen A11004), goat anti-rabbit 568 (1/1000; Invitrogen A11011), goat anti-mouse 647 (1/1000; Invitrogen A21235), and goat anti-rabbit 647 (1/1000; Invitrogen A21244). DNA stains used: Hoechst (1/1000; Molecular Probes H-3569). Apoptotic cells were detected in dissected guts using the TMR red In Situ Cell Death Detection Kit (Roche; 12156792910) following standard kit staining protocol. Briefly, guts were washed in PBS following secondary antibody then stained with 100 µl of TUNEL solution for 1 h at 37°C then washed twice with PBS. Intestines were mounted on slides using Fluoromount (Sigma F4680). Clonal analyses (Flip-out system) were proceeded with 10-day-old female adults (raised and aged at 18°C) that were transferred to 29°C for 3 days to induce esg-clones and transgenes expression. Flies were then kept at 25°C for 5 days, prior to dissections. Flies were transferred to fresh vials every 2-3 days. For every experiment, images were obtained of the posterior midgut region (R4/5) of the intestine with a spinning disk confocal microscope (Quorum WaveFX). PH3+ cells were counted through the entire midgut. For clonal analysis, clones were counted as groups of three of more GFP+ cells.

### DSS treatment
DSS was prepared by dissolving DSS (Sigma 42867) in a PBS 5% sucrose solution, filter sterilized and kept in the freezer for up to 2 weeks. Flies were raised on a 5% DSS solution unless stated otherwise. DSS vials were prepped by covering normal fly food with circular filter paper (Whatman, Grade 3, 23 mm, 1003-323) and adding 150 µl of DSS, or control (PBS 5% sucrose) solutions. Flies were flipped daily onto fresh DSS, or control solution for 48 h for all experiments except for DSS survival, which was over the course of 10 days.

### Lifespan
For longevity, 30 virgin females per vial were raised at 29°C and dead flies were counted every 1-3 days and vials were flipped three times per week to fresh standard food. For DSS survival experiments flies were placed on fresh 10% DSS daily for the course of the survival experiment and deaths were counted daily.

### Data visualization and statistical analysis
Figures were constructed using R (version 4.1.2) via R studio with easyggplot2 (version 1.0.0.9000) or ggplot2 (version 3.3.5). Statistical analysis was performed in R. Figures were assembled in Adobe Illustrator.

### Sample prep for single cell RNA sequencing
Preparation of single-cell intestinal suspension was made following previous methods (31, 44). Flies were raised for 10 days at 29°C then treated with 3% DSS or sucrose/PBS solution for 48 h. Batches of five *Drosophila* midguts were dissected at once then transferred to 1% BSA in DEPC treated PBS.

Biology Open

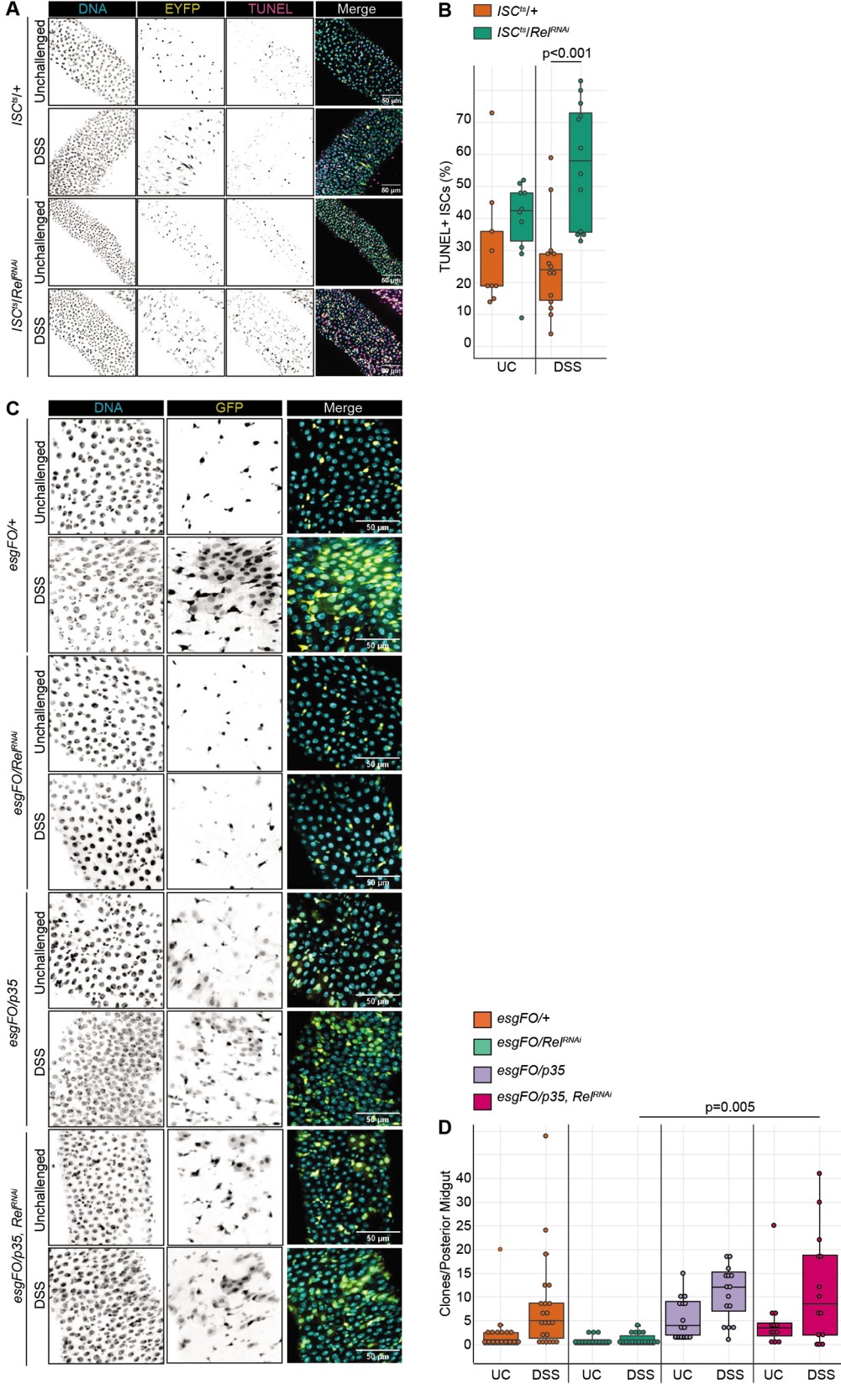

**Fig. 7. Rel promotes stem cell survival during an enteric challenge with DSS.** (A) DNA (cyan), GFP+ progenitors (yellow) and TUNEL-positive cells (magenta) in control or DSS-treated wild-type (*ISC^{ts}/+*) and Rel-deficient (*ISC^{ts}/Rel^{RNAi}*) flies. (B) TUNEL positive ISCs in control, and DSS-treated *ISC^{ts}/+* and *ISC^{ts}/Rel^{RNAi}* intestines. (C) DNA (cyan) and GFP-marked clones (yellow) generated by unchallenged or DSS-treated intestines of the indicated genotypes. We induced flip-out clones by incubating flies at 29°C, followed by 48 h treatment with DSS (passed into fresh vials after 24 h), with an additional 5 days of recovery at 25°C to allow daughter cell regeneration. (D) Quantification of clone numbers in the indicated genotypes and treatment groups. Significance for B and D were calculated using ANOVA followed by pairwise Tukey tests.

Once 30 midguts were obtained for each condition, they were transferred to a 1.5 ml tube with 200 µl of DEPC/PBS with 1 mg/ml Elastase (Sigma, E0258) and chopped into pieces with small dissecting scissors. After mechanical disruption, tubes were incubated at 27°C for 40 min with gentle pipetting every 10 min. 22 µl of 10%BSA in DEPC/PBS solution was added to stop the enzymatic disruption then cells were pelleted by spinning at 300 g for 15 min at 4°C. Cell pellet was resuspended in 200 µl of 0.04% BSA in DEPC/PBS then filtered through a 70 µm filter. Live cells were enriched using OptiPrep

Density Gradient Medium (Sigma, D1556). Filtered cells were mixed with 444 µl of 40% iodixanol (2:1 OptiPrep:0.04% BSA DEPC/PBS) then transferred to a 15 ml tube. Another 5.36 ml of 40% iodixanol was added and mixed. A 3 ml layer of 22% iodixanol was added on top then an additional 0.5 ml layer of 0.04% BAS in PBS/DEPC was added. Tubes were spun at 800 $g$ for 20 min at 20°C then the top interface containing live cells (∼500 µl) was collected. Live cells were diluted with 1 ml of 0.04% BSA in DEPC/PBS. Remaining iodixanol was removed by pelleting cells at 300 g for 10 min at 4°C and removing supernatant. Cell pellet was resuspended in remaining 0.04% BSA DEPC/PBS solution (∼40 µl) and cell counts and viability was determined using a hemocytometer. Cell viability was as follows: *ISCts/+* unchallenged=96%, *ISCts/+* DSS=95%, *ISCts/relRNAi* unchallenged=97%, *ISCts/relRNAi* DSS=86%. Libraries were generated using 10X Genomics Single-cell Transcriptome Library kit and sent to Novogene for sequencing.

## Bioinformatics

Raw sequencing data from Novogene was aligned to the *Drosophila* reference transcriptome (FlyBase, r6.30) using Cell Ranger v3.0 with the EYFP sequence appended to generate feature-barcode matrices. The resulting matrices were analyzed using Seurat (v4.1.0) in R. Cells with <200 or >3500 features and cells with >20% mitochondrial reads were removed to reduce number of low-quality cells or doublets. Expression values were normalized, and data clustering was performed at a resolution of 0.4 with 30 principal components. Clusters were identified using established markers and previous *Drosophila* intestine single-cell analysis (www. flyrnai.org/scRNA). Gene Ontology term analysis was performed using DAVID.

## Acknowledgements

*Drosophila* lines were kindly provided by Dr Lucy O'Brien and Dr Bruce Edgar (Stanford University and University of Utah, respectively). We acknowledge microscopy support from the Faculty of Medicine and Dentistry Imaging core; and support with single-cell library preparation from the Faculty of Medicine and Dentistry Advanced Cell Exploration core.

## Competing interests

The authors declare no competing or financial interests.

## Author contributions

Conceptualization: M.F., E.F.; Data curation: A.J., M.F.; Formal analysis: A.J., M.F., E.F.; Funding acquisition: E.F.; Investigation: A.J., M.F., M.S., E.F.; Methodology: A.J., M.F.; Project administration: E.F.; Supervision: E.F.; Visualization: A.J., M.F.; Writing – original draft: A.J., M.F., E.F.; Writing – review & editing: A.J., M.F., E.F.

## Funding

This work was supported by a grant from the Canadian Institute of Health Research (grant #PJT 159604). M.F. received funding through Alberta Innovates Graduate Student Scholarships and an NSERC PGS-D. Open Access funding provided by University of Alberta. Deposited in PMC for immediate release.

## Data and resource availability

Gene expression data is deposited on NCBI under the accession number PRJNA873108.

## Peer review history

The peer review history is available online at https://journals.biologists.com/bio/lookup/doi/10.1242/bio.062025.reviewer-comments.pdf.

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
