## [Peer Review File · Biology Open]

Stem cell-specific NF- κ B is required for stem cell survival and epithelial regeneration upon intestinal damage

Aurélia Joly, Meghan Ferguson, Minjeong Shin and Edan Foley

DOI: 10.1242/bio.062025

Editor: Tristan Rodriguez

Review timeline

Original submission:	19 February 2025
Editorial decision:	28 March 2025
First revision received:	26 June 2025
Accepted:	30 June 2025

Original submission

First decision letter

MS ID#: bio.062025

MS Title: Stem cell-specific NF- κ B is required for stem cell survival and epithelial regeneration upon intestinal damage

Authors: Aurélia Joly, Meghan Ferguson, Minjeong Shin and Edan Foley

I have now received all the referees reports on the above manuscript, and have reached a decision. The referees' comments are appended below, or you can access them online: please go to: .

As you will see from their reports, the referees recognise the potential of your work, but they also raise significant concerns about it. Given the nature of these concerns, I am afraid I have little choice other than to reject the paper at this stage.

Reviewer 1

In their paper, the authors use a combination of progenitor knockdown of Rel using RNAi and single cell seq analysis to conclude that stem cell specific NF- κ B is required for stem cell survival and regeneration epithelial regeneration upon intestinal damage. This paper is a follow up their previous paper in Stem Cell Reports (2022).

As it stands, for the reasons delineated below, I am not convinced that the phenotypes can be assigned to a requirement for Rel in the ISC. Indeed, in the Discussion, the authors write, "However, we cannot exclude the possibility that Rel controls survival via intermediaries." I am also concerned that this paper represents an incremental advance of what was demonstrated in the 2022 Stem Cell Reports paper.

Suggestions to improve the manuscript are as follows:

Other Major concerns

- 1) Figure 1C. The y-axis is labeled as PH3+ cells per midgut. But the text suggests the only the posterior midgut was analyzed.
- 2) Figure 1C. The graph suggests there are at most 10 PH3+ cells at 30 days. I seem to recall that most aging papers show a much more dramatic number.
- 3) Figure 1E. What do the authors mean by "per nucleus" in the figure legend.
- 4) "Combined, our data implicate intestinal Rel activity in progenitor cell proliferation and identity." The authors should avoid using sentences with vague conclusions. Write explicitly what

you found. I also do not see how the results from this section say anything about progenitor cell identity. The simplest interpretation is that Rel is required for ISC proliferation in aged adults.

5) Mutant clones of Rel and other IMD pathway members should be made in young and aged animals. The number of cells per clone (or some quantitative measure) and the clone composition (enterocytes, enteroendocrine cells) should then be determined.

6) "By day ten, more than 85% of ISCs/RelRNAi flies died, whereas 60% of control ISCs/+ flies remained alive (Supplementary Figure 2D), pointing to stem cell-specific roles for Rel in surviving enteric stress." It's unclear why the authors state the role is stem cell specific, especially since they claim Rel depletion did not affect ISC proliferation. A requirement for REL in progeny, including EBs and enterocytes that might inherit RNAi, could account for the phenotype. Furthermore, Gal80 in the ISCs line is only turned on after the Gal80 has time to be transcribed and translated. This delay might allow Rel to be made in the early enteroblast. Su(H)Gbe+-Gal4 may help. But it's worth noting that Rel knockdown only occurs in enteroblasts sometime after their specification.

7) I'm concerned with the use of Delta as an ISC marker (Supplementary Figure 2). Delta is often expressed in non ISCs in stressed/injured intestines.

8) Some quantitative measure of clone size should be used in Figure 1. Even something as simple as clones with <3 enterocytes vs those with >3 would suffice.

9) The small size of Rel flip out clones without a decrease in ISC proliferation after DSS suggests that Rel is required in ISC progeny and not the ISC.

10) Labial is a verified marker of Copper cells yet appears negative in Supplementary Figure 3B. Similar concern with Vha100-4

11) The transition from using esgts to ISCs should be explained explicitly in the text.

12) Diet can have profound effects on ISC proliferation and enterocyte differentiation. Many of the experiments (DSS, single cell seq) were done on sucrose only diets. Do the authors think this could affect the phenotypes or interpretations of results?

13) Esg flip out driving p35 would drive its expression in all cells of the clone.

14) Given that the IMD pathway is involved in immunity, it would be helpful to know what the effect of loss of Rel after damage in axenic flies.

Minor concerns

1) It seems peculiar to make the first two figures of the paper supplementary ones.

2) By day ten, more than 85% of ISCs/RelRNAi flies died, whereas 60% of control ISCs/+ flies remained alive (Supplementary Figure 2D), pointing to stem cell-specific roles for Rel in surviving enteric stress. Comparing flies died (85% vs 40%) or alive (15% vs 60%) between the two groups would make the cited sentence less awkward.

3) What is the difference between a middle enterocyte and a copper cell?

4) "By contrast". Not "in contrast."

5) References should be listed for flies used in the Methods section.

Reviewer 2

SUMMARY OF THE ADVANCE MADE IN THIS PAPER AND ITS POTENTIAL SIGNIFICANCE TO THE FIELD

In this manuscript, Joly et al. explore the role of the IMD pathway in stem cells and progenitors in response to chemical injury in the fly intestine. I think this is a good description of the role of Relish in progenitors to combat damage to DSS feeding. However, before this manuscript can be accepted, I would like the authors to address the following points in a revised version.

SUGGESTIONS TO AUTHORS

Major comments:

1. The organization of the manuscript is confusing. Figure 1 only appears on Page 8, 3 pages after Supplementary Figure 1. I think the supplementary Figure 1 should be included as a main Figure.

2. In Figure 1, the material methods do not explain how the experiment was carried out. Some details can be provided in the legend of the Figure. Especially in the case of clonal analysis across the manuscript, what were the kinetics of the heat shock, and how many hours/days was this done after the DSS feeding? This applies to Figure 3 as well.

3. In Supplementary Figure 1E, how were the Su(H) positive cell counts determined? Are these counts for individual guts or a specific region of the gut? Why are the basal counts at 30 days different between Supplementary Figure 1E and G?
4. In Supplementary Figures 1F and G, are the effects seen on enteroblasts at day 30 in RelishRNAi flies reversible with antibiotics?
5. Please show the survivals in Supplementary Figure 2D DSS-treated esgts/+ till the end of the curve where all DSS adults have died.
6. In Figure 3A, is there no significant difference between the WT and DSS feeding? Shouldn't DSS induce some apoptosis, as revealed by the TUNEL+ staining?
7. Are all the effects on decreased progenitors in the absence of Relish and increased apoptosis specific to DSS feeding? Have the authors tried other compounds like Bleomycin or a toxic organic compound like Paraquat?
8. The 'pre-enterocytes' identified in the 5% DSS of WT intestines deserve a separate Figure panel with their gene markers for the community to identify these signatures in their datasets.

Minor comments

1. On Page 7, line 5, a 'w' is repeated.
2. Information (catalog number) on the 10X Genomics library kit is missing.

Reviewer 3: SUMMARY OF THE ADVANCE MADE IN THIS PAPER AND ITS POTENTIAL SIGNIFICANCE TO THE FIELD

This study from the Foley lab characterizes the specific roles of Relish transcription factor in intestinal stem cells (ISC) for epithelial regeneration after DSS challenges. Using single cell transcriptomics, they show that ISC-specific depletion of Relish led to impaired ISC differentiation, the absence of a damage-induced pre-enterocyte population, and increased ISC apoptosis, in response to DSS damage. While this paper describes several interesting observations, I feel it generally lacks insights of how all the observations are mechanically linked. Extending on the following points should improve the paper.

SUGGESTIONS TO AUTHORS

Major comments

1. Relish in ISC differentiation

It is surprising to see such strong effect in ISC differentiation (absence of new ECs) by knocking down Relish during DSS challenge. Can this result be consolidated with an additional Relish-RNAi or RNAi lines targeting other IMD components? Better characterization of the differentiation defects should also be done by quantifying with the esgF/O system the numbers of new ECs generated in response to damage, rather than simply counting the number of clones (quite often it is difficult to separate individual clones).

Furthermore, the increase in ISC apoptosis upon Relish KD looks convincing, but how this is linked with the differentiation defects? Does Relish directly control the transcriptional program for damage-induced ISC differentiation or just protect ISC from cell death? The author should make this clear so that the role of Relish in ISCs can be properly appreciated. To this aim, checking the expression of markers/regulators of ISC differentiation upon Relish depletion can be done. Additionally, digging into the scRNA-seq data will likely give implications of what are acting downstream of Relish for ISC differentiation and/or prevention of ISC apoptosis.

2. The pre-EC population

This is an interesting finding of the paper that needs a bit extension. Ideally, the authors should find markers (based their scRNA-seq data) to visualize the pre-EC cells in the gut, and show they are induced by DSS damage in a Relish-dependent manner. This should increase the impact of the paper.

3. scRNA-seq data

I am a bit concerned about the quality of the scRNA-seq data. As you will see, essentially nearly all genes are downregulated upon Relish knockdown during DSS challenge (Fig 2F), but the majority of

genes are upregulated by knocking down Relish in basal condition (Fig S4F). Can the authors explain such strong bias of gene modulation by Relish in different conditions?

4. Microbial sensing in ISCs?

Repeating the key findings in germ-free condition is needed to clarify if ISC differentiation intrinsically controlled by Relish during DSS treatment requires microbial sensing in ISCs

Minor points

1. I suggest the authors stick to the term like ISC-intrinsic Relish function, rather than making claims related to ISC immunity. It is unclear if ISCs express AMPs and AMPs there have any role in gut immunity. ECs are known as the immune-competent cells that act to control gut bacteria.

2. Figure 2H: I do not see clear upregulation of cell death genes in the cluster of progenitor. These genes can be listed together with their levels of regulation by Relish.

3. introduce ISCs when it first appears.

4. There are better ways to show disrupted organization of intestinal epithelial cells (Fig S2A), e.g. with A142-GFP or bbg-GFP reporter.

5. please check for typos.

As IMD (acts) through the JNK and NF- κ B/Relish...

In contrast, whereas wildtype progenitors...

First revision

Author response to reviewers' comments

Reviewer 1: In their paper, the authors use a combination of progenitor knockdown of Rel using RNAi and single cell seq analysis to conclude that stem cell specific NF- κ B is required for stem cell survival and regeneration epithelial regeneration upon intestinal damage. This paper is a follow up their previous paper in Stem Cell Reports (2022).

As it stands, for the reasons delineated below, I am not convinced that the phenotypes can be assigned to a requirement for Rel in the ISC. Indeed, in the Discussion, the authors write, "However, we cannot exclude the possibility that Rel controls survival via intermediaries." I am also concerned that this paper represents an incremental advance of what was demonstrated in the 2022 Stem Cell Reports paper.

Suggestions to improve the manuscript are as follows:

Other Major concerns

1) Figure 1C. The y-axis is labeled as PH3+ cells per midgut. But the text suggests the only the posterior midgut was analyzed.

This suggestion is challenging. Aside from the fact that it appears to be a minor comment that can be readily addressed by correcting the text, Figure 1 does not have a panel C. We believe the reviewer means supplementary Figure 1C, as that panel does have PH3+ cell counts. However, there are no inconsistencies between text and y-axis in supplementary 1C. As the reviewer correctly points out the y-axis is labelled "PH3+ cells per midgut". The corresponding legend states (C) "PH3+ mitotic cells in *esg^{ts}/Rel^{RNAi}* and *esg^{ts}/+* midguts" and the accompanying text states "progenitor-restricted loss of *Rel* prevented age-dependent decline of epithelial organization (Supplementary Figure 1B) and resulted in significantly fewer ISC mitoses (Supplementary Figure 1C)." As the figure, legend, and text all overlap, we are unable to address this concern.

2) Figure 1C. The graph suggests there are at most 10 PH3+ cells at 30 days. I seem to recall that most aging papers show a much more dramatic number.

As above, there is no Figure 1C, so we will proceed with the assumption that this comment refers to Supplementary Figure 1C. Here, it appears the reviewer is expressing concerns based on their recollections. As no backing evidence or citations are provided, this is also a somewhat difficult suggestion to process. In our experience, there is no standard number for how many PH3+ cells one should expect in a 30d old intestine. Intestinal mitoses are influenced by many, features that include, but are certainly not limited to, genetic background of the flies, food, humidity,

endogenous microbiota, frequency of flipping, number of flies per vial, sex, mating status, lighting conditions, and handling. For that reason, many groups will note lab-specific mitotic indices. However, there is a consensus within the field that mitotic frequency rises as flies age, and we clearly reproduce this phenotype (see for example Supplementary figure 1C).

3) Figure 1E. What do the authors mean by "per nucleus" in the figure legend.

In Supplementary Figure 1E, we had used "nucleus" as a proxy for cell count. We have now rectified the legend to state that panel E shows "Percentage of intestinal epithelial cells that are GFP+ in 30-day old *esg^{ts}/Rel^{RNAi}* and *esg^{ts}/+* midguts."

4) "Combined, our data implicate intestinal Rel activity in progenitor cell proliferation and identity." The authors should avoid using sentences with vague conclusions. Write explicitly what you found. I also do not see how the results from this section say anything about progenitor cell identity. The simplest interpretation is that Rel is required for ISC proliferation in aged adults.

We have removed the words "and identity" to address this major concern.

5) Mutant clones of Rel and other IMD pathway members should be made in young and aged animals. The number of cells per clone (or some quantitative measure) and the clone composition (enterocytes, enteroendocrine cells) should then be determined.

We agree that deeper characterization of clones deficient for IMD in future studies is interesting. Unfortunately, it's not immediately clear how we should apply this to the manuscript under consideration. In our brief, focussed manuscript, we present experimental data to support the hypothesis that "Stem cell-specific NF- κ B is required for stem cell survival and epithelial regeneration upon intestinal damage" (seemanuscript title). We feel that all presented data support our hypothesis, and we believe these findings are of interest to the field.

6) "By day ten, more than 85% of ISCTs/RelRNAi flies died, whereas 60% of control ISCTs/+ flies remained alive (Supplementary Figure 2D), pointing to stem cell-specific roles for Rel in surviving enteric stress." It's unclear why the authors state the role is stem cell specific, especially since they claim Rel depletion did not affect ISC proliferation. A requirement for REL in progeny, including EBs and enterocytes that might inherit RNAi, could account for the phenotype. Furthermore, Gal80 in the ISCTs line is only turned on after the Gal80 has time to be transcribed and translated. This delay might allow Rel to be made in the early enteroblast. Su(H)Gbe+-Gal4 may help. But it's worth noting that Rel knockdown with only occur in enteroblasts sometime after their specification.

Reviewer 1 raises an interesting point about the fly line used for manipulation of transgene expression in the adult intestinal stem cell. For stem cell-specific transgene expression, we used the *w ; esg-GAL4,UAS-2xYFP;Su(H)GBE-GAL80,tub-GAL80^{ts} (ISC^{ts})* line. In this line, the *esg-GAL4 ; tub-GAL80^{ts}* elements allows temperature-induced transgene expression in EYFP-marked progenitors, while *Su(H)GBE-GAL80* constitutively suppresses transgene expression in enteroblasts, leading to stem cell-restricted transgene expression in EYFP-marked stem cells.

First described by the Jasper group over a decade ago (PMID: 25945494), the *ISC^{ts}* line used in this study is inarguably the most widely used line for stem cell-restricted control of gene expression in the adult gut, and has contributed to numerous advances in the field, including uncovering links between EGFP signaling and mitochondrial biogenesis (PMID: 35896119); establishing the involvement of polycomb genes in aging (PMID: 33724181); identifying regionalized patterns of gene expression in the gut (PMID: 26146076); revealing links between autophagy and proliferation (PMID: 31006650); and establishing the involvement of TOR in cell fate decisions (PMID: 38335286). We are unaware of any concerns about transient transgene expression in early enteroblasts for these papers, or for any of the many other papers that use the *ISC^{ts}* line.

To directly determine the efficacy of the *ISC^{ts}* line as a stem cell-specific driver, we examined our single cell gene expression data for expression of *EYFP*, *DI*, and *Su(H)* in the progenitors of *ISCTs/+* flies. As expected, we observed no overlap for the stem cell marker *DI*, and the enteroblast marker *Su(H)*. In addition, we found that 52% of all *DI+* cells expressed *EYFP*, confirming effective transgene expression in the stem cells of *ISCTs/+* flies. Notably, we did not detect *EYFP* expression

in any *Su(H)*⁺ cell, further supporting the established use of this line for modification of transgene expression in the adult intestinal stem cell. However, as our work and the many cited studies detailed above do not formally exclude possible transgene expression in very early enteroblasts, we have modified the relevant text to adopt a more cautious tone. The revised text now says: “By day ten, more than 85% of *ISC^{ts}/Rel^{RNAi}* flies died, whereas only 40% of *ISC^{ts}/+* flies had died (Figure 2D), suggesting that *ISC^{ts}*-mediated depletion of *Rel* impacts the ability of adult flies to survive challenges with DSS. As a caveat, we note that we cannot formally exclude the possibility that the *ISC^{ts}/Rel^{RNAi}* fly line also impacts *Rel* expression in early enteroblasts.”

7) I'm concerned with the use of Delta as an ISC marker (Supplementary Figure 2). Delta is often expressed in non ISCs in stressed/injured intestines

Our use of Delta as an ISC marker is consistent with published literature that Delta is a reliable stem cell marker, including in DSS-treated flies. For example, the following text is reproduced from **Tissue damage- induced intestinal stem cell division in Drosophila by the Ip group** (PMID: 19128792):

“We performed a series of experiments to determine the cell-fate and stem cell properties after DSS feeding. Double staining for Delta and *Su(H)Gbe-lacZ* revealed that there was no overlap before or after DSS feeding (Figures 2D-2I, arrows), demonstrating that ISC and enteroblast fates are preserved. By another double- staining experiment, we found that all the phospho-H3-positive cells had Delta staining (Figures 2J-2O), demonstrating that after DSS feeding, ISC retained the ability to proliferate and contained a high level of Delta. Therefore, the asymmetry, cell-fate determination, and known properties of ISC are not affected during the experimental period of DSS feeding.”

8) Some quantitative measure of clone size should be used in Figure 1. Even something as simple as clones with <3 enterocytes vs those with >3 would suffice.

We agree with reviewer one. For this experiment, we defined clones as groups of three or more cells, and all clones were counted in regions R4 and R5 of the adult midgut. We have updated the figure legend and materials and methods section to reflect this.

The revised legends now state: “Quantification of clones of three or more cells in regions R4 or R5 of the posterior midguts of flies with the indicated genotypes and treatments. Significance calculated using ANOVA followed by pairwise Tukey tests.”

The revised methods (Immunofluorescence section) now reads: “For clonal analysis, clones were counted as groups of three or more GFP⁺ cells.”

9) The small size of Rel flip out clones without a decrease in ISC proliferation after DSS suggests that Rel is required in ISC progeny and not the ISC.

As we observe elevated stem cell death in DSS challenged *Rel*-deficient stem cells, and we find that expression of the pan caspase inhibitor p35 effectively restores epithelial regeneration, we feel our data continue to support the hypothesis that *Rel* promotes cell survival in DSS-challenged intestines.

10) Labial is a verified marker of Copper cells yet appears negative in Supplementary Figure 3B. Similar concern with Vha100-4

All cell identities are based on a previous publication from the Perrimon group (**A cell atlas of the adult *Drosophila* midgut, PNAS 2020**; PMID: 31915294). In their study, the authors used *lab* and *Vha100-4* as a middle enterocyte (mEC) marker. The relevant text is reproduced below:

“Another cluster, middle ECs (mEC), mapped to the middle region of the midgut based on the regional expression of the *lab* transcription factor (*SI Appendix, Fig. S2E*) (14, 29-31), as well as *Vha100-4*, a component of Vacuolar H⁺ ATPase required for acid generation (32).

To ensure consistency between our work and their widely cited study (166 citations by June 2025), we adopted the same naming convention.

11) The transition from using *esgts* to ISCs should be explained explicitly in the text.

This was an important suggestion that we have now addressed in the revised manuscript. The revised results section now states: “Since the *esg* driver allows for the expression of transgenes in both ISCs and enteroblasts, we wanted to focus on roles of *Rel* specifically in ISCs. To do this we

employed the *ISC^{ts}* fly line (*w ; esg-GAL4,UAS-2xYFP;Su(H)GBE-GAL80,tub-GAL80^{ts}*) to knock down *Rel* specifically in ISCs, then monitored fly survival in response to DSS.”

12) Diet can have profound effects on ISC proliferation and enterocyte differentiation. Many of the experiments (DSS, single cell seq) were done on sucrose only diets. Do the authors think this could affect the phenotypes or interpretations of results?

We concur with the reviewer that food is an interesting variable to consider when examining gut phenotypes. However, in our case, none of the experiments was done on sucrose only meals. As per the materials and methods, all flies were fed normal fly food. For challenge experiments, the food was covered with a thin filter paper that was soaked in DSS or sucrose solutions, which allows the underlying food to saturate the disc. This approach ensures flies ingest DSS, without limiting access to nutrients in the food. Confirming that the flies are not starved under this widely used experimental paradigm, Figure 2D show that *ISC^{ts/+}* and *ISC^{ts/Rel^{RNAi}}* flies survive a minimum of ten days under these conditions without any mortality.

13) Esg flip out driving p35 would drive its expression in all cells of the clone.

Reviewer one is correct, and we have updated the text accordingly. The revised text now reads:

“To determine if failed epithelial repair occurs in *ISC^{ts/Rel^{RNAi}}* midguts due to excess ISC death, we tracked midgut regeneration in control flies (*esg^{ts} F/O/+*); flies that lack *Rel* within the progenitor compartment (*esg^{ts} F/O/Rel^{RNAi}*); flies that express the anti-apoptotic p35 protein in progenitors (*esg^{ts} F/O/p35*), and flies that express p35 in *Rel*-deficient progenitors (*esg^{ts} F/O/Rel^{RNAi}, p35*). These fly lines express the respective transgenes in all GFP-marker progenitors and their mature progeny.”

14) Given that the IMD pathway is involved in immunity, it would be helpful to know what the effect of loss of *Rel* after damage in axenic flies.

We concur and had raised a very similar point in our Discussion:

“If paracellular leakage of bacterial patterns initiates immune responses in progenitors, it would be of interest to test how defined bacterial communities, including absence of gut bacteria, modify stem cell survival and barrier regeneration after a DSS challenge. With a simple, manipulable microbiome, and straightforward infection protocols, flies are an excellent model to ask how hosts, commensals, and pathogens interact to shape stress-dependent epithelial regeneration^{10,20,56,71}.”

However, as the reviewer would doubtlessly appreciate accurate comparisons between conventional, axenic, and gnotobiotic flies are highly complex undertakings that go far beyond the scope of a brief report, such as this manuscript. For that reason, we prefer to retain an exclusive focus on how *Rel*-deficient stem cells react to DSS treatment in the presence of a conventional microbiome. We feel this work has value to those interested in stem cell biology and hope that it might inspire detailed examinations of microbial contributions in future studies.

Minor concerns

1) It seems peculiar to make the first two figures of the paper supplementary ones.

We concur with this assessment and struggled throughout the writing of this manuscript to come up with an effective organization of the figures. Our initial arrangement was primarily the result of the limited number of figures allowed by a report format in Development. As Biology Open allows a greater number of figures, we have now removed the designation “supplementary” from all figures, to produce a seven-figure manuscript that flows more naturally with the text.

2) By day ten, more than 85% of *ISCs/RelRNAi* flies died, whereas 60% of control *ISCs/+* flies remained alive (Supplementary Figure 2D), pointing to stem cell-specific roles for *Rel* in surviving enteric stress. Comparing flies died (85% vs 40%) or alive (15% vs 60%) between the two groups would make the cited sentence less awkward.

We concur with this recommendation and have adapted the text accordingly. The revised text now states: “By day ten, more than 85% of *ISC^{ts/Rel^{RNAi}}* flies died, whereas only 40% of control

ISC^{ts}/+ flies had died”.

3) What is the difference between a middle enterocyte and a copper cell?

As stated above, all cell identities are based on “A cell atlas of the adult *Drosophila* midgut” from the Perrimon group. The relevant text is reproduced below:

“Another cluster, middle ECs (mEC), mapped to the middle region of the midgut based on the regional expression of the *lab* transcription factor (*SI Appendix*, Fig. S2E) (14, 29-31), as well as *Vha100-4*, a component of Vacuolar H⁺ ATPase required for acid generation (32). Another cluster mapped to copper and iron cells based on the expression of *PGRP-SC2*, and a number of metal ion binding proteins, such as *MtnA*, *MtnB*, *MtnC*, and *MtnD*.”

4) "By contrast". Not "in contrast."

Reviewer one is incorrect. As per the Merriam-Webster English dictionary (<https://www.merriam-webster.com/dictionary/by%2Fin%20contrast>), “by contrast” and “in contrast”, are synonymous idioms: **by/in contrast idiom**: when compared to another: when looked at or thought about in relation to similar objects or people to set off dissimilar qualities

5) References should be listed for flies used in the Methods section.

This is an important oversight that we have corrected in the revised version.

Reviewer 2: SUMMARY OF THE ADVANCE MADE IN THIS PAPER AND ITS POTENTIAL SIGNIFICANCE TO THE FIELD

In this manuscript, Joly et al. explore the role of the IMD pathway in stem cells and progenitors in response to chemical injury in the fly intestine. I think this is a good description of the role of Relish in progenitors to combat damage to DSS feeding. However, before this manuscript can be accepted, I would like the authors to address the following points in a revised version.

SUGGESTIONS TO AUTHORS

Major comments:

1. The organization of the manuscript is confusing. Figure 1 only appears on Page 8, 3 pages after Supplementary Figure 1. I think the supplementary Figure 1 should be included as a main Figure.

This aligns with a concern raised by reviewer one, and with our initial challenges fitting our data into a three- figure format report. In our revised manuscript, we have removed the “supplementary” designation from all figures, which allows text and data to integrate much more seamlessly.

2. In Figure 1, the material methods do not explain how the experiment was carried out. Some details can be provided in the legend of the Figure. Especially in the case of clonal analysis across the manuscript, what were the kinetics of the heat shock, and how many hours/days was this done after the DSS feeding? This applies to Figure 3 as well.

We have updated the corresponding legends to address this important point. The legends now read: “We induced flip-out clones by incubating flies at 29°C, followed by 48h treatment with DSS (passed into fresh vials after 24h), with an additional 5 days of recovery at 25°C to allow daughter cell regeneration.”.

3. In Supplementary Figure 1E, how were the Su(H) positive cell counts determined? Are these counts for individual guts or a specific region of the gut? Why are the basal counts at 30 days different between Supplementary Figure 1E and G?

We believe this might be a misunderstanding on the part of reviewer two. S1E does not show counts for Su(H)⁺ cells. S1E shows “Percentage of intestinal epithelial cells that are GFP⁺ in 30-day old *esg^{ts}/Rel^{RNAi}* and *esg^{ts}/+* midguts” (quote from the accompanying legend). As *esg* marks stem cells and enteroblasts, this driver does not provide Su(H) positive cell counts. Su(H) positive cells were counted in supplementary Figure 1G where we counted “Proportion of Su(H)⁺ enterocyte precursors within the indicated progenitor pools”. In each instance each data point corresponds to a single intestine, and all images were taken from the R4/R5 posterior midgut region (From the materials and methods: “For every experiment, images were obtained of the posterior midgut region (R4/5) of the intestine with a spinning disk confocal microscope (Quorum WaveFX).”

4. In Supplementary Figures 1F and G, are the effects seen on enteroblasts at day 30 in RelishRNAi flies reversible with antibiotics?

This is an interesting question, and connects to a point made by reviewer one, and elaborated on by us in our discussion - does the microbiota affect the interplay between Relish, epithelial stress, and stem cell survival. We concur that all related questions are interesting and share the reviewer's enthusiasm for this area of research. However, the purpose of our manuscript was to determine how Relish deficiency impacted the response of a conventionally raised fly to challenges with the widely used colitogenic agent DSS. We feel this work should be of interest to the field and argue that meaningful dissections of microbial contribution to Rel-DSS interactions go far beyond the scope of our current, focussed report.

5. Please show the survivals in Supplementary Figure 2D DSS-treated esgts/+ till the end of the curve where all DSS adults have died.

Supplementary figure 2D shows the entirety of the survival curve. Ten days is consistent with DSS treatments for experimental animals. As most DSS-exposed Rel-deficient flies had died at that point, and all control flies were still alive, extending the experiment would have had no impact on the conclusions.

6. In Figure 3A, is there no significant difference between the WT and DSS feeding? Shouldn't DSS induce some apoptosis, as revealed by the TUNEL+ staining? Like reviewer 2, we had initially anticipated elevated TUNEL+ staining in DSS-exposed WT flies. Unfortunately, this area appears relatively understudied within the literature, and so we have minimum available comparisons for our findings. We note that our data reflect the % TUNEL+ ISCs. As DSS-treated flies go through a burst of ISC proliferation, our results suggest that there is an increase in the absolute numbers of dying ISCs, but that the rate of death is matched by the rate of proliferation, resulting in no apparent change in the overall percentage of dying ISCs.

7. Are all the effects on decreased progenitors in the absence of Relish and increased apoptosis specific to DSS feeding? Have the authors tried other compounds like Bleomycin or a toxic organic compound like Paraquat?

This is an interesting question that we agree merits following up in future work. Looking at Bleomycin, paraquat, or additional challenges such as bacterial pathogens, radiation exposure, etc. would likely uncover additional roles for Rel in stem cell responses to stress. However, our goal with this manuscript was to specifically characterize the relationship between Rel deficiency and stem cell responses to a DSS challenge. We felt this focussed approach had merit, as it allowed us to provide preliminary mechanistic links between Rel function and stem cell survival during exposure to a widely used colitogenic agent.

8. The 'pre-enterocytes' identified in the 5% DSS of WT intestines deserve a separate Figure panel with their gene markers for the community to identify these signatures in their datasets.

This matches a comment made by reviewer three. In our revised manuscript, we have added a new panel (panel C) that highlight markers associated with the pre-enterocyte cluster.

Minor comments

1 On Page 7, line 5, a 'w' is repeated.

We have rectified this error.

2. Information (catalog number) on the 10X Genomics library kit is missing.

Unfortunately, we do not have this number. The single cell data was generated more than two years ago, and we do not have a catalog number for the kit.

Reviewer 3: SUMMARY OF THE ADVANCE MADE IN THIS PAPER AND ITS POTENTIAL SIGNIFICANCE TO THE FIELD

This study from the Foley lab characterizes the specific roles of Relish transcription factor in intestinal stem cells (ISC) for epithelial regeneration after DSS challenges. Using single cell transcriptomics, they show that ISC-specific depletion of Relish led to impaired ISC differentiation, the absence of a damage-induced pre-enterocyte population, and increased ISC apoptosis, in response to DSS damage. While this paper describes several interesting

observations, I feel it generally lacks insights of how all the observations are mechanically linked. Extending on the following points should improve the paper.

SUGGESTIONS TO AUTHORS

Major comments

1. Relish in ISC differentiation

It is surprising to see such strong effect in ISC differentiation (absence of new ECs) by knocking down Relish during DSS challenge. Can this result be consolidated with an additional Relish-RNAi or RNAi lines targeting other IMD components? Better characterization of the differentiation defects should also be done by quantifying with the *esgF/O* system the numbers of new ECs generated in response to damage, rather than simply counting the number of clones (quite often it is difficult to separate individual clones).

Our single cell gene expression data confirmed the efficacy and cell-type specificity of the *Rel* RNAi line used in this study. Figure 2E confirms that exposure of wildtype flies to DSS induces *Rel*-responsive transcripts (*pirk*, *PGRP-SD*, *PGRP-LB*) without altering *Rel* expression, and Figure 2H confirms progenitor-specific knock down of *Rel*, *PGRP-LB*, and *PGRP-LF* in *ISC^{ts}/Rel^{RNAi}* flies raised under identical conditions. We concur with reviewer three that the possibility of individual clones merging can lead to an under calculation of the actual number of clones generated in the epithelium. However, we would also like to point out that we only observe large clones in wildtype (*esgF/O/+*) flies treated with DSS (Figure 1), and never in DSS-treated *Rel*-deficient (*esgF/O/Rel^{RNAi}*) counterparts (Figure 1). As a result, the interpretation that loss of *Rel* prevents generation of clones remains fully supported by the data.

Furthermore, the increase in ISC apoptosis upon Relish KD looks convincing, but how this is linked with the differentiation defects? Does Relish directly control the transcriptional program for damage-induced ISC differentiation or just protect ISC from cell death? The author should make this clear so that the role of Relish in ISCs can be properly appreciated. To this aim, checking the expression of markers/regulators of ISC differentiation upon Relish depletion can be done. Additionally, digging into the scRNA-seq data will likely give implications of what are acting downstream of Relish for ISC differentiation and/or prevention of ISC apoptosis.

We concur that this is an interesting question raised by the findings in our manuscript. In our discussion, we reviewed the downstream events from *Rel* engagement in considerable detail:

“How does *Rel* modify stem cell survival? As *Rel* has broad effects on gene expression within the midgut, including genes linked to stress responses and development³², it is possible that *Rel* directly modifies expression of genes involved in cell survival. However, we cannot exclude the possibility that *Rel* controls survival via intermediaries. Moving forward, it will be of interest to test if interactions between *Rel* and EGF, Notch or JAK-STAT are important for DSS-dependent control of stem cell viability. Beyond, EGF, Notch, and JAK-STAT, we feel it will be pertinent to test the tumor suppressor BMP pathway as a potential mediator of *Rel*-dependent stem cell survival. BMP regulates epithelial patterning^{28,38,72-77}; DSS activates the BMP pathway in progenitors (Figure 3D); and *Vibrio cholerae* engages a *Rel*-BMP axis to blocks ISC proliferation⁷⁸, suggesting possible interactions between *Rel* and BMP in the regulation of cell viability after exposure to DSS. Moving forward, reviewer three is correct that mining our publicly accessible datasets can provide novel insights into progenitor cell-specific responses to *Rel* deficiency, in the presence or absence of a DSS challenge. As this reviewer, and the editor can doubtlessly appreciate, meaningful studies in these directions require an extensive set of new experiments that are unrelated to the manuscript under consideration. For that reason, we prefer to keep the focus of this manuscript exclusively on exploring the links between our transcriptomic data and stem cell survival.

2. The pre-EC population

This is an interesting finding of the paper that needs a bit extension. Ideally, the authors should find markers (based their scRNA-seq data) to visualize the pre-EC cells in the gut, and show they are induced by DSS damage in a Relish-dependent manner. This should increase the impact of the paper.

This suggestion is similar to a comment raised by reviewer 2. In our revised manuscript, we clarify that pre- enterocytes are primarily marked by expression of genes linked to metabolism (as one

would expect for enterocytes), and we present a new panel (Figure 2C) to highlight three prominent pre-enterocyte markers.

3. scRNA-seq data

I am a bit concerned about the quality of the scRNA-seq data. As you will see, essentially nearly all genes are downregulated upon Relish knockdown during DSS challenge (Fig 2F), but the majority of genes are upregulated by knocking down Relish in basal condition (Fig S4F). Can the authors explain such strong bias of gene modulation by Relish in different conditions?

We concur that our work shows broad effects of Rel on gene expression. However, we note that our work matches an earlier demonstration that Rel is a prominent regulator of gene expression within the fly gut (PMID: 24865556). The reviewer is correct that our work points to distinct impacts of Rel deficiency on homeostatic intestines relative to DSS-treated intestines, however we view this finding as consistent with finding from multiple model organisms that DSS-dependent damage of the intestinal epithelium has substantial effects on gene expression within the gut.

4. Microbial sensing in ISCs?

Repeating the key findings in germ-free condition is needed to clarify if ISC differentiation intrinsically controlled by Relish during DSS treatment requires microbial sensing in ISCs

Like reviewers one and two, reviewer three raises the interesting point that gut microbes may have noteworthy effects on Rel-dependent responses to DSS. As stated earlier, and as pointed out in our discussion, we agree with this point, and feel it would be of considerable interest to use a simple model like the fly to systematically address the effects of gut bacteria on Rel signaling in stem cells. However, we would once again like to emphasize that such a study goes far beyond the scope of this focussed report. Most groups find that their flies associate with five to ten bacterial taxa, each of which has distinct effects on gut physiology. We strongly feel that an accurate assessment of this microbial community on host responses to DSS requires a detailed set of genetic, genomic, and quantitative studies that ultimately belong in follow-up work.

Minor points

1. I suggest the authors stick to the term like ISC-intrinsic Relish function, rather than making claims related to ISC immunity. It is unclear if ISCs express AMPs and AMPs there have any role in gut immunity. ECs are known as the immune-competent cells that act to control gut bacteria.

We agree and have changed the manuscript accordingly.

2. Figure 2H: I do not see clear upregulation of cell death genes in the cluster of progenitor. These genes can be listed together with their levels of regulation by Relish.

This was an error on our part. We had used an incorrect color scale for the heatmap in the corresponding figure. We have since rectified this error, and use the corresponding text to highlight canonical apoptotic regulators that are sensitive to Rel deficiency

3. introduce ISCs when it first appears.

We have followed through on this suggestion.

4. There are better ways to show disrupted organization of intestinal epithelial cells (Fig S2A), e.g. with A142-GPF or bbg-GFP reporter.

We concur that the suggested reporters are very useful for showing cytoskeletal disruptions in DSS-treated flies. However, we feel that the images and quantitative data presented in Figure 2 adequately document the intestinal damage experienced by flies treated with DSS.

5. please check for typos.

As IMD (acts) through the JNK and NF-kB/Relish...

In contrast, wWhereas wildtype progenitors...

We have checked for typos throughout and modified the manuscript accordingly.

Second decision letter

MS ID#: bio.062025

MS Title: Stem cell-specific NF- κ B is required for stem cell survival and epithelial regeneration upon intestinal damage

Authors: Aurélie Joly, Meghan Ferguson, Minjeong Shin and Edan Foley

I am happy to tell you that your manuscript has been accepted for publication in Biology Open, pending our standard publication integrity checks. It was accepted on 30 June 2025.